# DRIVING BY THE RULES: A BENCHMARK FOR INTEGRATING TRAFFIC SIGN REGULATIONS INTO VECTORIZED HD MAP

## ABSTRACT

Ensuring adherence to traffic sign regulations is essential for both human and autonomous vehicle navigation. While current benchmark datasets concentrate on lane perception or basic traffic sign recognition, they often overlook the intricate task of integrating these regulations into lane operations. Addressing this gap, we introduce **MapDR**, a novel dataset designed for the extraction of **D**riving **R**ules from traffic signs and their association with vectorized, locally perceived HD **Maps**. MapDR features over 10,000 annotated video clips that capture the intricate correlation between traffic sign regulations and lanes. We define two pivotal sub-tasks: 1) **Rule Extraction from Traffic Sign**, which accurately deciphers regulatory instructions, and 2) **Rule-Lane Correspondence Reasoning**, which aligns these rules with their respective lanes. Built upon this benchmark, we provide a multimodal solution that offers a strong baseline for advancing autonomous driving technologies. It fills a critical gap in the integration of traffic sign rules, contributing to the development of reliable autonomous navigation systems.

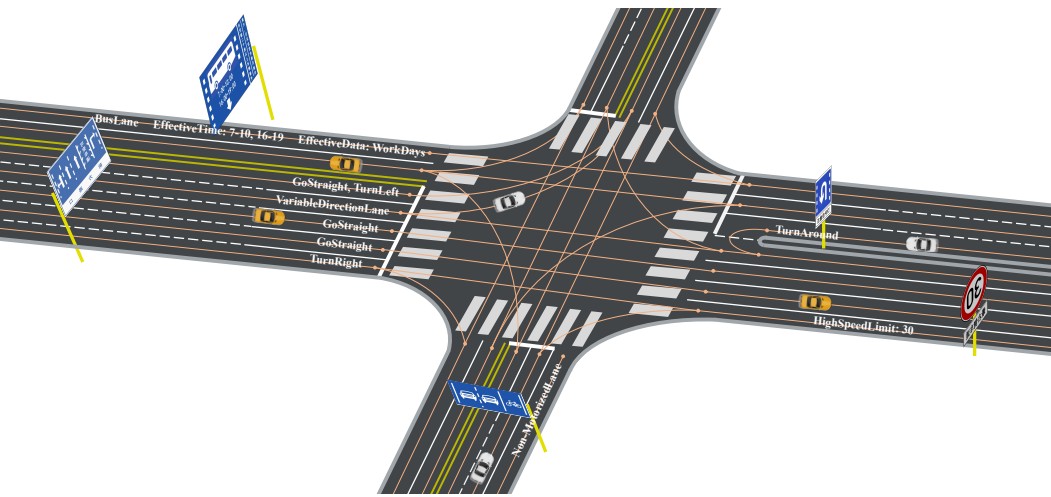

Figure 1: **MapDR Overview and Motivation**. For safe autonomous driving, accurate interpretation of lanes and traffic signs is crucial, ensuring vehicles maintain proper positioning and follow driving rules. This figure illustrates an intersection scene where extracted traffic sign rules are integrated into the corresponding lanes on the HD map.

## 1 INTRODUCTION

The emergence of autonomous vehicles and intelligent transportation systems has highlighted the critical need for accurate and reliable navigational data. High-Definition (HD) maps [1], with their

---

[1]The HD map discussed in this paper refers to a local vectorized map constructed through online perception by autonomous vehicles.

detailed representation of the road environment, have become indispensable for these advanced systems. Traffic signs, as the visual language of the road, are essential for conveying driving rules such as speed limits, lane usage restrictions, and right-of-way rules. For autonomous vehicles, accurate recognition and interpretation of these signs are not just advantageous but essential for safe and compliant operation on public roads. However, current online HD map construction for autonomous driving mainly focuses on accurately depicting the types and positions of map elements in BEV space using point sequences, neglecting the driving rules conveyed by traffic signs and their relation to lanes.

Beyond mere recognition, effective autonomous navigation demands a deeper integration of traffic signs into the vehicle's HD map, as depicted in Figure 1. The conventional researches of sign detection and classification Behrendt et al. (2017); Stallkamp et al. (2012); Fregin et al. (2018); Zhu et al. (2016); Yu et al. (2020), which often rely on single labels, are inadequate for capturing the detailed requirements of lane-level driving rules. A single traffic sign often represents multiple rules applicable to various lanes, each with distinct attributes such as lane direction and speed limitations. The challenge lies in binding these lane-level rules to the corresponding lanes within the HD map. Achieving this level of integration is essential for developing HD map that can robustly support autonomous driving.

Despite the critical role that traffic sign integration plays in autonomous driving, there has been a noticeable lack of focused research in this area. The CTSU dataset Guo et al. (2021), for instance, takes an initial step by encoding traffic signs in $\{key : value\}$ pairs, yet it does not effectively link the semantic content of signs to specific lanes. Other efforts, such as OpenLaneV2 Wang et al. (2023) and VTKGG Guo et al. (2023) have attempted to establish connections between traffic signs and lanes. However, they have not fully addressed the structural interpretation of the multifaceted attributes of lane-level rules.

To address this gap, we introduce **MapDR**, the first dataset specifically designed for driving rules extraction from traffic signs and association with vectorized HD maps. MapDR provides an extensive collection of over $10,000$ video clips that explore the correlation between lanes and driving rules extracted from traffic signs. For more details on the proposed dataset, please refer to Section 4.

MapDR introduces two innovative sub-tasks aimed at bolstering research in this domain: **1) Rule Extraction from Traffic Sign**: This sub-task is dedicated to developing algorithms that can extract specific lane-level rules from traffic signs, including their attributes and the lanes to which they apply. It is an essential step for understanding the intricate details of traffic signs and their navigational implications. **2) Rule-Lane Correspondence Reasoning**: This sub-task focuses on establishing a precise relationship between the extracted rules and the corresponding lanes in the HD maps. This process is vital for autonomous systems to accurately contextualize and apply lane-level rules to their driving path. For detailed descriptions of the proposed tasks and metrics, please refer to Section 3.

Based on the proposed tasks and dataset, we leverage multimodal models to design a solution that **integrating traffic sign regulations into vectorized HD maps**. This provides a strong baseline for future research work. We hope to inspire more researchers to focus on this task and drive the development of related industries.

To sum up, our contributions are as follows:

- For the first time, we introduce the task of extracting lane-level rules from traffic signs and integrating them into vectorized HD maps. Additionally, we present the MapDR dataset and specific metrics for benchmarking this task.

- MapDR comprises an extensive collection of images from three representative Chinese cities, captured over a quarter year at various times of the day. This dataset includes over $10,000$ video clips, at least $400,000$ front-view images, and more than $18,000$ lane-level rules. All annotations are carefully validated, with all data newly collected.

- We present Vision-Language Encoder (VLE) and Map Element Encoder (MEE) to extract and interact features from image, text, and vector data, integrating lane-level rules into vectorized HD maps and providing an effective baseline for future researches.

Table 1: **Comparison of the existing datasets.** "Sign" and "Lane" denote whether the dataset focus on traffic signs and lanes. Only those annotated with formatted ("Fmt.") rules and the correspondence ("Corr.") between rules and lanes can form driving rules. "Clip" represents whether the data is organized in the form of video clips. "$*$" denotes that these samples are not newly collected and are built upon the previous dataset.

| Dataset | Sign | Lane | Driving Rules | | Number of Samples | | | Year |
| | | | Fmt. | Corr. | Image | Clip | Region | |
| --- | --- | --- | --- | --- | --- | --- | --- | --- |
| nuScenes Caesar et al. (2020) | | ✓ | | | $1400K$ | $1K$ | Worldwide | 2019 |
| Argoverse2 Wilson et al. (2021) | | ✓ | | | $2100K$ | $1K$ | USA | 2021 |
| CTSU Guo et al. (2021) | ✓ | | | | $5K$ | / | China | 2021 |
| OpenLane Chen et al. (2022) | | ✓ | | | $200K^*$ | $1K^*$ | Worldwide | 2022 |
| RS10K Guo et al. (2023) | ✓ | | | ✓ | $10K$ | / | China | 2023 |
| OpenLaneV2 Wang et al. (2023) | ✓ | ✓ | | ✓ | $466K^*$ | $2K^*$ | Worldwide | 2023 |
| **MapDR(ours)** | ✓ | ✓ | ✓ | ✓ | **$400K$** | **$10K$** | **China** | **2024** |

## 2 RELATED WORK

### 2.1 HD MAP CONSTRUCTION

HD maps construction have seen significant advancements, with a focus on traffic element perception, including lane detection and traffic sign recognition Wilson et al. (2021); Huang et al. (2020); Caesar et al. (2020); Gu et al. (2019); Behrendt et al. (2017); Stallkamp et al. (2012); Yu et al. (2020); Fregin et al. (2018); Zhu et al. (2016). The shift towards BEV perception and vectorization for end-to-end HD maps construction has gained traction Wilson et al. (2021); Caesar et al. (2020); Chen et al. (2022). Notable works include HDMapNet, which aggregates semantic segmentation results Li et al. (2022b), LSS Philion & Fidler (2020) estimates depth to transfer image features to BEV features, while VectorMapNet Liu et al. (2023c) is the first end-to-end framework for sequential vector point prediction to generate HD maps without post-processing. MapTR Liao et al. (2023a) and its enhanced version, MapTRv2 Liao et al. (2023b), introduced a unified permutation-equivalent modeling approach and extended it to a general framework supporting centerline learning and 3D map construction. However, these efforts have largely overlooked the integration of traffic sign rules into HD maps.

### 2.2 TRAFFIC ELEMENT ASSOCIATION

Traffic element association aims to link elements like traffic signs with lanes. As demonstrated in Table 1, CTSU has initiated internal elements association to describe traffic sign in $\{key : value\}$ form, however lacking both generalization of driving rules from description and lane association Guo et al. (2021). VTKGG Guo et al. (2023) propose to utilize a graph model for connectivity but also lacks structured expression of driving rules for motion planning and requires complex integration into HD maps, which is typically expressed in the BEV space. OpenLaneV2 Wang et al. (2023) advances BEV space association but is constrained by single-label classification, making it insufficient for signs with multiple rules, which are common in real scenarios. Recent MLLM-based benchmarks Marcu et al. (2023); Qian et al. (2024); Sachdeva et al. (2024); Sima et al. (2023); Cao et al. (2024) for autonomous driving, such as MAPLM Cao et al. (2024), prioritize end-to-end motion planning over precise rule extraction from traffic sign, lacking evaluation for rule reasoning. MapDR addresses this gap by focusing on traffic sign rule extraction and lane association.

### 2.3 VISION-LANGUAGE MODELS

Vision-Language Models (VLMs) facilitates multimodal applications by learning joint representations of vision and language data. Visual Question Answer (VQA) tasks provide answers to image-related questions Antol et al. (2015), while Visual Information Extraction (VIE) tasks extract structured information from visual and textual data Antol et al. (2015); Xu et al. (2020; 2021); Huang et al. (2022). In Autonomous Driving (AD), VLMs are increasingly used for comprehensive traffic scene understanding and decision-making. The field has seen various approaches, including using transformers Vaswani et al. (2017) for joint encoding Kim et al. (2021); Huang et al. (2022), excelling

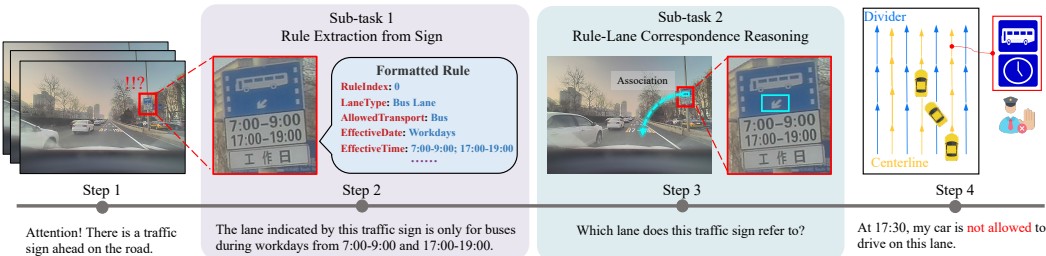

Figure 2: **Overview of the task.** $Step\ 1 \sim Step\ 4$ shows a case of driving by the rules. $Step\ 2$ and $Step\ 3$ demonstrates the specific role of two sub-tasks, respectively.

at multimodal information interaction, and independent encoders for different modalities Radford et al. (2021); Jia et al. (2021) that are proficient in multimodal retrieval. Cross-modal representation methods Li et al. (2021); Yu et al. (2022) combine these advantages, and the latest LLM-based research Li et al. (2023); Liu et al. (2023b;a; 2024) has achieved state-of-the-art results in various multimodal tasks. Nowadays, an increasing number of methods are leveraging LLMs to achieve impressive results, with works like DriveLLM Cui et al. (2024) showing significant potential in AD. However, addressing hallucination Bai et al. (2024) remains the most crucial aspect for LLM-based approaches.

## 3 TASK DEFINITION : INTEGRATING TRAFFIC SIGN REGULATIONS INTO HD MAPS

The ability to discern rules from traffic signs and to associate them with specific lanes is pivotal for autonomous navigation. As depicted in Figure 2, traffic signs are primary indicators of lane-level rules. Our proposed task involves two core sub-tasks: **1) Extracting lane-level rules from traffic signs**, and **2) Establishing correspondence between these rules and centerlines**. Generally, vehicles follow the center of lanes , *i.e.*, centerlines, to drive on the road Wang et al. (2023). Therefore, we use centerlines to represent lanes. This approach mirrors human drivers' instinct to observe traffic signs and then relate the indicated rules to the lanes they govern.

### 3.1 RULE EXTRACTION FROM TRAFFIC SIGN

As shown in $Step\ 2$ of Figure 2, this task involves extracting multiple rules $R = \{r_i\}_{i=1}^m$ from a series of image sequences $X = \{x_i\}_{i=1}^n$, where $m$ is the number of rules and $n$ is the number of frames. Each rule $r_i$ is a set of pre-defined properties in $\{key : value\}$ pairs. The rule extraction model, denoted as $\mathcal{M}$, can be expressed as $R = \mathcal{M}(X)$. To facilitate this challenging task, existing algorithm results for sign detection and OCR, represented as $B$ and $T$ respectively, can be utilized, making the rule extraction process $R = \mathcal{M}(X, [B], [T])$, $[\cdot]$ indicates optional input.

### 3.2 RULE-LANE CORRESPONDENCE REASONING

As shown in $Step\ 3$ of Figure 2, the reasoning process establishes the correspondence between centerlines $L = \{l_i\}_{i=1}^k$ and all rules $R$, where $k$ is the number of centerlines. We denote the correspondence reasoning model as $\mathcal{T}$, and this process can be described as $E = \mathcal{T}(R, L)$, where $E \in \{0,1\}^{m \times k}$ and the element $E_{ij}$ in the $i$-th row and $j$-th column of matrix $E$ represents the corresponding status between $r_i$ and $l_j$. The final reasoning result forms a bipartite graph $G = (R \cup L, E)$, which means corresponding relationships only exist between rules and centerlines .

## 4 THE MAPDR DATASET & BENCHMARK

We introduce the MapDR dataset, meticulously annotated with traffic sign regulations and their correspondences to lanes, as shown in Figure 3. The dataset encompasses a diverse range of scenarios, weather conditions, and traffic situations, with over $10,000$ traffic scene segments, $18,000$ driving

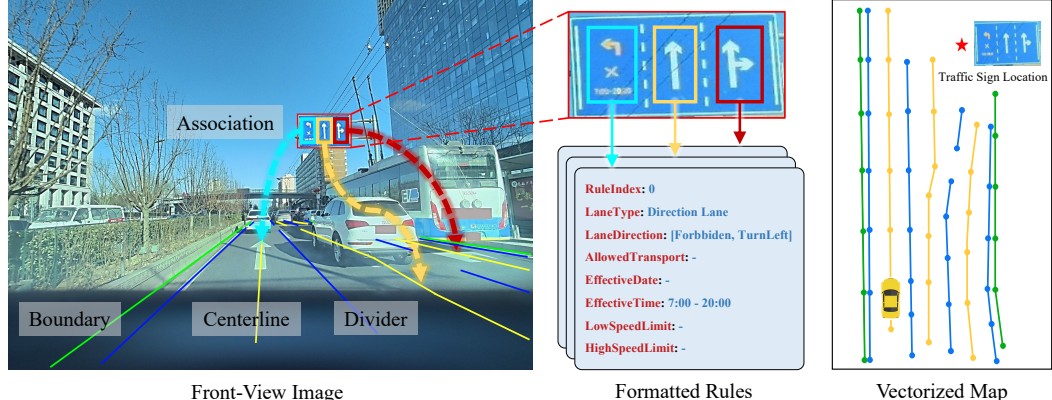

Figure 3: **Visualization of dataset demo.** Multiple lane-level rules of a single traffic sign are annotated in $\{key : value\}$ format. Directed lines indicate the correspondence between rules and particular centerlines.

rules, and $400,000$ images. Traffic signs typically have varying textual descriptions, text layouts, and positions on the road, which add complexity to the task.

The majority of the data originates from Beijing and Shanghai, with additional scenes from Guangzhou. Figure 4 illustrates the geographic spread and variety of traffic signs. The dataset reflects a natural long-tail distribution, with a prevalence of bus and direction lanes and a scarcity of tidalflow lanes. We primarily focus on traffic signs that indicate lane-level rules, collected from cities with the most complex and diverse traffic scenarios in China, ensuring realistic and representative data. All images have undergone privacy and safety processing to obscure license plates and faces. More comprehensive statistic of dataset and case demonstrations can be found in appendix H.

### 4.1 RAW DATA & ANNOTATION

**Raw Data.** MapDR is collected from real-world traffic scenes, each scene segment (video clip) captures front-view images within a $100m \times 100m$ area centered on the traffic sign, with a consistent resolution of $1920 \times 1240$. Each clip contains 30 to 60 frames, captured at 1 frame every 2 meters, ensuring consistent spatial intervals. Each video clip focuses on a single traffic sign and provides its position in 3D space. Camera intrinsics and poses are provided for each frame, and coordinates for each clip are transformed to distinct ENU systems. For safety and privacy, the reference point is not provided. All vectors of local map in the target area are provided as 3D point lists, generated using

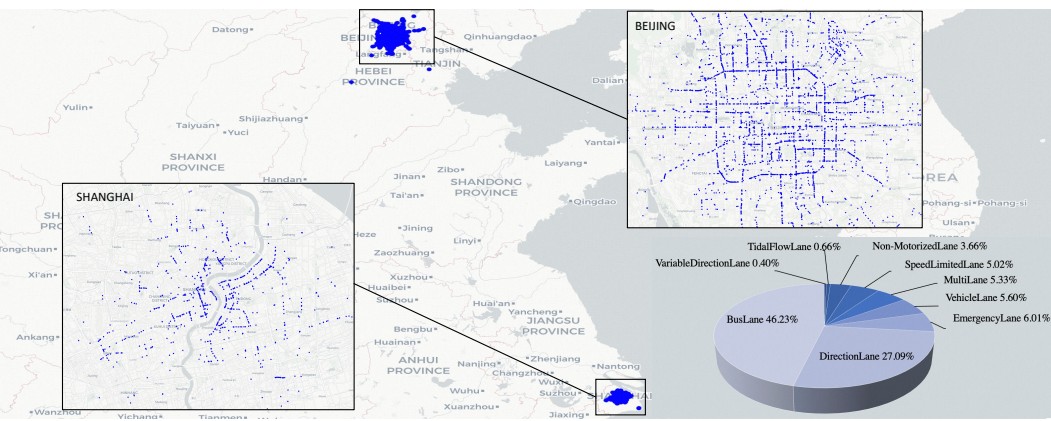

Figure 4: **Geographic location distribution of the collected traffic signs and proportions of various lane types represented in all signs.** The geographic distribution is visualized based on OpenStreetMap osm.

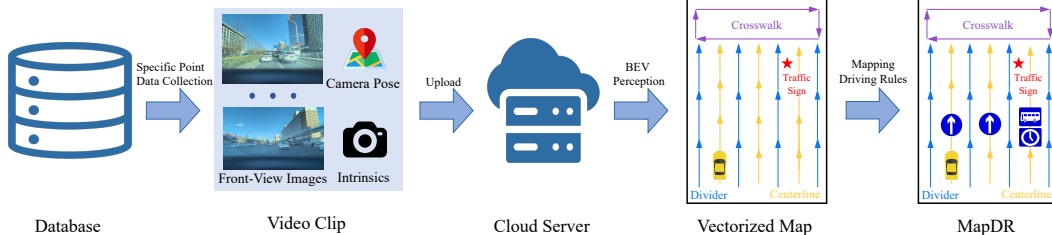

Figure 5: **Pipeline of dataset production.** The location of traffic signs are sampled from existing database then front-view images of each sign are newly collected. Vectorized map is processed in cloud sever. Finally formatted rules and correspondence between rules and centerlines are annotated and organized as MapDR.

our algorithm similar to MapTRv2 Liao et al. (2023b). Each lane vector has a type, such as divider, centerline, crosswalk, or boundary. For example, the centerline is defined as $L = \{l_i\}_{i=1}^k$, where each vector $l_i$ is composed of multiple 3D points $l_i = [p_1, \ldots, p_n]$, and $p_j = (x_j, y_j, z_j)$ represents the coordinates of the current point. The pipeline of dataset production is illustrated in Figure 5, and detailed data acquisition and annotation procedures can be found in the appendix F.

**Formatted Rules.** Each video clip may contain multiple lane-level rules, denoted as $R$. Each rule is expressed by symbols and text on the sign, requiring interpretation. As shown in Figure 3, each rule $r_i$ comprises 8 predefined properties in the form of $\{key : value\}$ pairs. We enclose the symbols and texts denoting each distinct rule on traffic signs with polygons and project them into 3D space as $P_i = [p_1, \ldots, p_n]$, where $n$ varies. Researchers can optionally use this information to facilitate rule extraction.

**Correspondence between Rules & Lanes.** Based on formatted rules $R$ and centerlines $L$, corresponding centerlines of each rule are annotated as shown in Figure 3. Therefore correspondence between rules and centerlines can be formed as a bipartite graph $G = (R \cup L, E)$, where $E \in \{0, 1\}^{|R| \times |L|}$ and the positive edges only exist between $R$ and $L$ as demonstrated in Section 3.2. Specifically, $E_{ij} = 1$ represents that vehicle driving on the lane with centerline $l_j$ should follow the driving rule $r_i$.

### 4.2 EVALUATION METRICS

We evaluated the two sub-tasks separately and then assessed the overall task performance. Methods are supposed to be ranked according to the overall $AP$.

**Rule Extraction (R.E.).** Given the ground truth $R$ and predicted rules $\hat{R}$, we propose to calculate the $Precision$ ($P_{R.E.}$) and $Recall$ ($R_{R.E.}$) to evaluate the capability of rules extraction as defined in Equation equation 1, where $\hat{r}_i = r_i$ represents all the properties are predicted correctly.

$$P_{R.E.} = \frac{|\hat{R} \cap R|}{|\hat{R}|} \qquad R_{R.E.} = \frac{|\hat{R} \cap R|}{|R|} \qquad (1)$$

**Correspondence Reasoning (C.R.).** Given the ground truth of correspondence bipartite graph $G = (R \cup L, E)$ and predicted graph $\hat{G} = (R \cup L, \hat{E})$, we propose to calculate $Precision$ ($P_{C.R.}$) and $Recall$ ($R_{C.R.}$) of edge set $E$ to evaluate the capability of correspondence reasoning individually. Metrics are defined as Equation equation 2.

$$P_{C.R.} = \frac{|\hat{E} \cap E|}{|\hat{E}|} \qquad R_{C.R.} = \frac{|\hat{E} \cap E|}{|E|} \qquad (2)$$

**Overall.** To evaluate the entire task, capability of both sub-tasks should be considered jointly. Therefore the predicted results are supposed to be the combination of two sub-tasks. Given the predicted

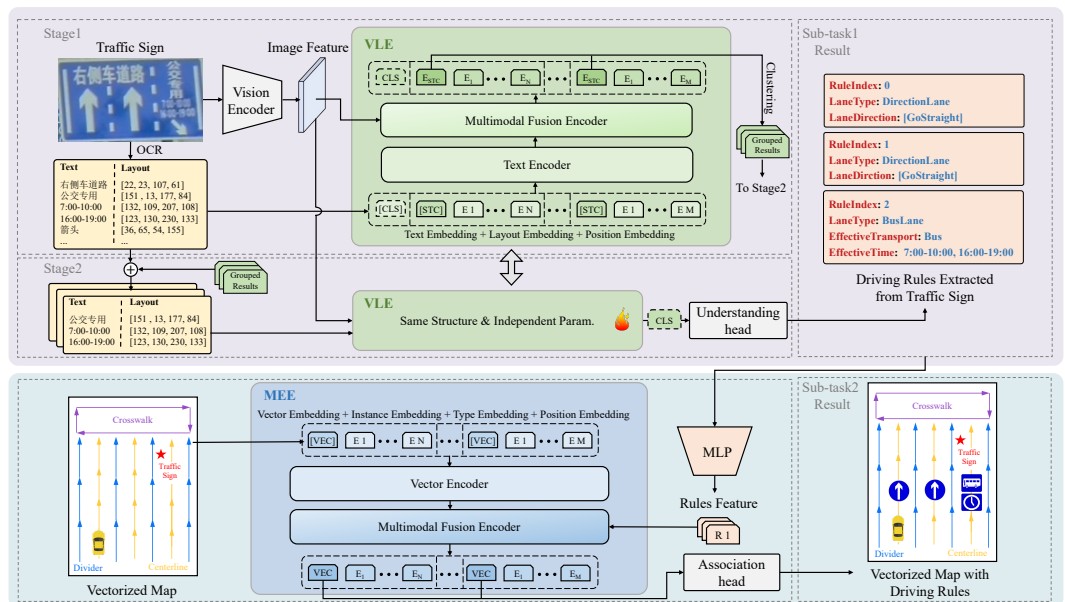

Figure 6: **Overview of the proposed method.** Entire approach can be divided into two main parts: **Rule Extraction from Traffic Sign** (top) and **Rule-Lane Correspondence Reasoning** (bottom). Rule Extraction model consists of two sequential stages with the same structure VLE but unshared parameters, and the training procedure is independent.

rules, correspondence should be reasoned between $\hat{R}$ and $L$ which means the prediction of entire task is $\hat{G} = (\hat{R} \cup L, \hat{E})$ and the ground truth is consistent $G = (R \cup L, E)$. We evaluate $Precision$ ($P_{all}$) and $Recall$ ($R_{all}$) using the sub-graph $G^s$, where $G^s = \{g_{ij}^s\}_{i=1,j=1}^{m,k}, g_{ij}^s = (\{r_i, l_j\}, e_{ij})$. In set of sub-graph $G^s$, $m$ is the number of rules, and $k$ is the number of centerlines. Furthermore, we propose the $average\ precision$ ($AP$) for the final benchmark ranking. Metrics are defined in Equation equation 3, $AP$ score is the area under the precision-recall curve, where $p$ and $r$ denote $P_{all}$ and $R_{all}$ respectively. We provide an example of calculating the $Overall$ metrics in appendix I.

$$P_{all} = \frac{|\hat{G}^s \cap G^s|}{|\hat{G}^s|} \qquad R_{all} = \frac{|\hat{G}^s \cap G^s|}{|G^s|} \qquad AP = \int_0^1 p(r)dr \qquad (3)$$

# 5 A BASELINE METHOD FOR MAPDR

To tackle the multimodal information interaction involving images, texts, and vectors, we develop a **Vision-Language Encoder (VLE)** and a **Map Element Encoder (MEE)**. The following sections detail their structures and applications, as well as the experimental results on MapDR.

## 5.1 ARCHITECTURE

**Vision-Language Encoder.** Inspired by vision-language frameworks Li et al. (2021; 2022a); Radford et al. (2021); Kim et al. (2021); Bao et al. (2022), we designed a vision-language fusion model named VLE, following Li et al. (2021). As shown in Figure 6, VLE uses ViT-b16 Dosovitskiy et al. (2021) as the vision encoder, with the text encoder and multimodal fusion encoder each consisting of $L$ transformer layers Vaswani et al. (2017). Each layer of the fusion encoder includes a cross-attention module for fusion Li et al. (2021). In practice, distinct rules are represented by varying numbers of symbols and texts, as shown in the OCR results in Figure 6. To address the challenge of representing variable-length input as fixed-length features, we introduce a `[CLS]` token for an entire rule and several `[STC]` tokens for sentence-level representation. The specific usage of these tokens is detailed in 5.2. Furthermore, we incorporate inter-instance and intra-instance attention mechanisms Liao et al. (2023b) to enhance model performance by capturing interactions

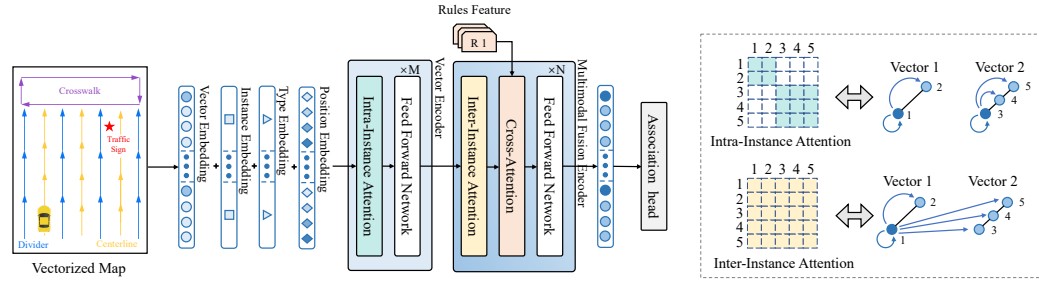

Figure 7: **Structure of MEE.** MEE serves as correspondence reasoning model. Learnable embeddings are introduced within input to enhance the representing capacity of vector types. inter & intra-instance attention mechanisms facilitate to capture the relationships and independence of individual vectors.

and independence between and within sentences. In addition to content, layout captures the relative positions of symbols and texts, offering important semantic meaning. To leverage this, we encode the layout using the method from Tancik et al. (2020) and the relative positions of characters as position embedding following Devlin et al. (2019). As shown in VLE in Figure 6, text embedding, layout embedding, and position embedding together form the input of the text encoder.

**Map Element Encoder.** Vectors can be represented as sequences of points, similar to words in sentences. Inspired by this, we designed MEE akin to language models Devlin et al. (2019). The MEE employs $M$ transformer layers for vector encoding and $N$ cross-attention layers for multimodal fusion. Utilizing the method from Tancik et al. (2020), points of each vector are embedded as point embedding. To achieve a fixed-length representation, we add [VEC] tokens as the first token of each vector, similar to [STC] tokens in the VLE. We also introduce learnable type embedding for vector types, learnable instance embedding to distinguish vector instances, and position embedding from Devlin et al. (2019) to encode the relative positions of multiple points within a vector. These embedding are aggregated as the input of vector encoder, as shown in Figure 7. In addition, we employ inter-instance and intra-instance attention mechanisms Liao et al. (2023b) to prioritize interactions within vectors over interactions between vectors, as depicted in the dashed box on the right side of Figure 7. The [VEC] token in output serves as fused feature of rules and vectors, enabling the final prediction of their relationships through association head.

## 5.2 IMPLEMENTATION

We utilize VLE and MEE as backbones to integrate multiple modalities and address these two sub-tasks. The specific procedures are detailed as follows:

**Rule Extraction from Traffic sign.** To clarify the objectives of model, we first *cluster symbols and texts into groups*. As shown in the upper part of Figure 6, the VLE is used to encode OCR results and images. By calculating the cosine similarity between [STC] tokens, different symbols and texts are clustered into groups. This process is supervised by contrastive loss during training. Next, using grouped OCR results as text input and maintaining the VLE structure, we *extract lane-level rules*. We employ a multi-classification head (understanding head) for the [CLS] token to predict the corresponding value for each attribute of the rules. This process allows us to express all rules inside a traffic sign as $\{key : value\}$ pairs.

**Rule-Lane Correspondence Reasoning.** MEE is designed for vector encoding and interaction with rules. Each formatted rule is mapped to an embedding through MLP and fused with vector features in the fusion encoder, as shown in the lower part of Figure 6. We add a binary classification head after each [VEC] token to determine the relationship between the current centerline and rule.

## 5.3 EXPERIMENT

**Setups.** The dataset is split into $train$ and $test$ sets in the ratio of $9 : 1$. $L = 6$ in VLE and $M = 2, N = 2$ in MEE. Input images are resized to $256 \times 256$ and the feature dimension is 768

Table 2: **Evaluation of the full pipeline.** VLE and MEE without any introduced technique serve as the baseline. Note that "$*$" denotes models can not converge in the setting.

| Model | R.E. | | C.R. | | Overall | | |
|---|---|---|---|---|---|---|---|
| | $P_{R.E.}(\%)$ | $R_{R.E.}(\%)$ | $P_{C.R.}(\%)$ | $R_{C.R.}(\%)$ | $P_{all}(\%)$ | $R_{all}(\%)$ | $AP(\%)$ |
| Baseline | 75.78 | 57.56 | $*$ | $*$ | $*$ | $*$ | $*$ |
| **VLE+MEE** | **76.67** | **74.54** | **78.05** | **82.16** | **63.35** | **67.37** | **44.60** |

Table 3: **Evaluation of sub-tasks.** Left: Rule Extraction, Right: Correspondence Reasoning. "Attn." indicates intra & inter-instance attention mechanisms. "Layout" refers to the text layout applied in VLE. "In.E." and "Ty.E." denotes instance and type embedding in MEE, respectively.

| VLE | | $P_{R.E.}(\%)$ | $R_{R.E.}(\%)$ |
|---|---|---|---|
| Attn. | Layout | | |
| ✗ | ✗ | 75.78 | 57.56 |
| ✓ | ✗ | 76.86 | 71.75 |
| ✓ | ✓ | **76.67** | **74.54** |

| MEE | | | $P_{C.R.}(\%)$ | $R_{C.R.}(\%)$ |
|---|---|---|---|---|
| Attn. | In.E. | Ty.E. | | |
| ✗ | ✗ | ✗ | $*$ | $*$ |
| ✓ | ✗ | ✗ | 68.91 | 71.39 |
| ✓ | ✓ | ✗ | 69.68 | 72.76 |
| ✓ | ✓ | ✓ | **78.05** | **82.16** |

with consistent 12 attention heads. We initialize VLE with pre-trained weights of DeiT Touvron et al. (2021) and BERT Devlin et al. (2019) while MEE is trained from scratch. The training procedure runs 50 and 120 epochs for VLE and MEE, respectively. All training employ $lr = 1e-4$, $wd = 0.02$ with AdamW Loshchilov & Hutter (2019) optimizer and cosine scheduler Loshchilov & Hutter (2017). More details can be found in the appendix J.

**Results.** We make minimal modifications to ALBEF Li et al. (2021) and BERT Devlin et al. (2019) to adapt them to our task, and we use this as our baseline. As shown in Table 2, the baseline method failed to converge during the correspondence reasoning procedure, resulting in no statistics evaluation. Table 3 indicates the attention mechanisms significantly improve $R_{R.E.}$, while layout of text brings marginal improvement. For the correspondence reasoning sub-task, the attention mechanisms enables MEE to converge. Instance embedding slightly improves $P_{C.R.}$ and $R_{C.R.}$, while type embedding significantly enhances both, indicating that vector types help the model establish rule-lane correspondence. The separate evaluation results of all lane types can also be found in appendix G

**Qualitative results of MLLMs.** We qualitatively evaluated the performance of existing MLLMs on the tasks of rule extraction and correspondence reasoning using a subset of MapDR. Specific details and results of the evaluation method are provided in Appendix K. The main conclusion of the evaluation shows that MLLMs understand traffic signs to a certain extent but lack spatial association capability. This indicates that MLLMs have tremendous potential, but still require careful design and optimization to adapt to this task. The findings further underscore the necessity of the modeling approach we have proposed, as it facilitates a more profound understanding of the task.

## 6 CONCLUSION

We introduce MapDR, a dataset with more than $10,000$ video clips, over $400,000$ images, and at least $18,000$ driving rules. This work defines the task of integrating traffic sign regulations into vectorized HD map, proposes a viable solution and establishes an effective baseline. With the emergence of MLLMs, we will explore their potential to tackle this complex comprehending task in future work.

**Limitation.** In our dataset, we do not consider the impact of dynamic elements, such as traffic lights, on driving rules, as these scenarios have already been discussed in previous works like OpenLaneV2 Wang et al. (2023). Instead, we focus on the impact of lane-level rules on driving, a topic often overlooked in previous datasets. In the future, we plan to incorporate these dynamic elements to create a more comprehensive dataset.

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

## A  APPENDIX OVERVIEW

Our appendix encompass author statements, licensing, dataset access, dataset analysis, and the implementation details of benchmark results to ensure reproducibility. Additionally, we offer dataset documentation in adherence to the Datasheet format Gebru et al. (2021), which covers details such as data distribution, maintenance plan, composition, collection, and other pertinent information.

## B  AUTHOR STATEMENT

We bear all responsibilities for licensing, distributing, and maintaining our dataset.

## C  LICENSING

The proposed dataset MapDR is under the CC BY-NC-SA 4.0 license, while the evaluation code is under the Apache License 2.0.

## D  DATASHEET

### D.1  MOTIVATION

**For what purpose was the dataset created?**    Autonomous driving not only requires attention to the vehicle's trajectory but also to traffic regulations. However, in the online-constructed vectorized HD maps, traffic regulations are often overlooked. Therefore, we propose this dataset to integrate lane-level regulations into the vectorized HD maps. These regulations can serve as navigation data for both human drivers and autonomous vehicles, and are crucial for driving behavior.

### D.2  DISTRIBUTION

**Will the dataset be distributed to third parties outside of the entity (e.g., company, institution, organization) on behalf of which the dataset was created?**    Yes, the dataset is open to public.

**How will the dataset be distributed (e.g., tarball on website, API, GitHub)?**    The dataset will be made public on *Tianchi* or *ModelScope*, while the evaluation code will be publicly released on *GitHub*.

### D.3  MAINTENANCE

**Is there an erratum?**    No. We will make a statement if there is any error are found in the future, we will release errata on the main web page for the dataset.

**Will the dataset be updated (e.g., to correct labeling errors, add new instances, delete instances)?**
Yes, the dataset will be updated as necessary to ensure accuracy, and announcements will be made accordingly. These updates will be posted on the dataset's webpage on *Tianchi* or *ModelScope*.

**Will older versions of the dataset continue to be supported/hosted/maintained?**    Yes, older versions of the dataset will continue to be maintained and hosted.

### D.4  COMPOSITION

**What do the instances that comprise the dataset represent?**    An instance of the dataset consists of three main parts: a video clip, basic information, and annotation. The video clip comprises at least 30 continuous front-view image frames, with one frame captured every 2 meters to ensure uniform spatial distribution. Basic information of each clip is presented in the form of a JSON file, including the locations of traffic sign, all lane vectors, camera intrinsic parameters, and the camera poses for each frame. Annotation is also organized in JSON format, containing multiple driving rules. Each rule consists of a set of properties in $\{key : value\}$ format, along with the index of each centerline

associated. All coordinates are transferred to the ENU coordinate systems, consistent within each segment but distinct between segments. For safety and privacy reasons, reference points are not provided.

**How many instances are there in total (of each type, if appropriate)?** MapDR is composed of $10,000$ newly collected traffic scenes with over $400,000$ front-view images, containing more than $18,000$ lane-level driving rules.

**Are relationships between individual instances made explicit?** The frames in a single video clip are continuous in time with a uniform spatial distribution. All video clips are collected among different time periods with consistent capture equipment and vehicles

**Are there recommended data splits (e.g., training, development/validation, testing)?** We have partitioned the dataset into two distinct splits: training and testing.

**Is the dataset self-contained, or does it link to or otherwise rely on external resources?** MapDR is totally newly collected and self-contained. Front-view images are captured and all the vectors are generated by our vectorized algorithm. All driving rules and correspondence are manually annotated.

### D.5 COLLECTION PROCESS

**Who was involved in the data collection process (e.g., students, crowdworkers, contractors) and how were they compensated (e.g., how much were crowdworkers paid)?** Based on our HD map annotation scheme and annotation team, we have provided high-quality annotations with the help of experienced annotators and multiple validation stages.

### D.6 USE

**What (other) tasks could the dataset be used for?** MapDR focus on the primary task of integrating driving rules from traffic signs to vectorized HD maps, which can be divided into two distinct sub-tasks: rule extraction and rule-lane correspondence reasoning. Researchers can also adapt to other traffic scene tasks.

## E ACCESS TO MAPDR

Due to the sensitive nature of the dataset, which involves geographical location information, **the full dataset is under review FOR NOW, and will be released in the camera-ready version**. During the review phase, we provide reviewers with a subset demonstration of MapDR, consisting of 180 video clips containing all types of lanes, to showcase the characteristics of this dataset.

### E.1 URL

**FOR NOW** Reviewers can download a subset of MapDR from URL below. Full dataset is under review and will be published in camera-ready.

- `https://drive.google.com/file/d/18wCZOWrysJJp8NQ-Pi03Xcz8_06nxZ1s/view?usp=sharing`

### E.2 EVALUATION CODE

We provide source code for sub-tasks and overall metric evaluation on MapDR. The evaluation code is available at the following URL link.

- `https://drive.google.com/file/d/13KVcwHd_6qj-q_92IjA1XnGhD971v_Kx/view?usp=sharing`

# F  DATASET PRODUCTION

## F.1  DATA PRODUCTION PIPELINE

**Data Collection.**    Search and Retrieval: We use out database to locate the GPS coordinates of traffic signs, utilizing both text-based and image-based retrieval methods. Route Planning: Our path planning algorithm is employed to design data collection routes. Vehicles equipped with data collection devices gather raw data, including images, camera parameters, and pose information, which are then uploaded to the cloud. Data Processing:

**Vectorization.**    In the cloud, BEV (Bird's Eye View) perception algorithms are applied to generate vectorized local HD maps. Key point detection and matching algorithms are used to recover the 3D positions of traffic signs.

**Rule Extraction.**    For each set of multiple image frames containing traffic signs, the most representative frame is selected for rule extraction by annotators. Vectorized map results are provided for annotating rule-lane associations. All captured images and the projection of vectorized maps in these images are included as reference material to enhance annotation accuracy.

## F.2  ANNOTATION PROCESS

**Rule Identification.**    Annotators identify the number of rules on each traffic sign and group related text information corresponding to each rule.

**Annotation Creation.**    A json file is created with eight properties that annotators fill based on their interpretation of the rules.

**Vector Association.**    Each rule is associated with the vector ID corresponding to its location on the vectorized map. Unique IDs are assigned to all vectors.

**Quality Assurance.**    Quality inspection procedures are implemented to ensure the accuracy of annotations. This includes a thorough review and rework process to correct any discrepancies.

# G  ANALYSIS OF MAPDR

**Data&Label Composition.**    MapDR is organized into video clips, with each clip focusing on a single traffic sign. The raw data and annotation are provided as JSON files. Table 6 demonstrates the composition of raw data. The demo is as shown in Listing 1. The 3D spatial location of the traffic sign is provided by 4 points represented as *traffic_board_pose*. Vectors and their types are also provided. Additionally, camera intrinsics and pose for each frame are provided to facilitate vector visualization. Note that all coordinates have been transferred to relative ENU coordinate systems which is consistent within a clip. Considering safety and privacy, the reference point is not provided. Table 7 shows the details of annotation. The demo is as shown in Listing 2. All pre-defined properties of driving rules are illustrated. The corresponding centerlines of each rule are annotated by the vector index. As mentioned in main submission, spatial location of the symbols and texts which represent the particular rules, referred to as semantic groups, is also provided. Researchers can optionally utilize this information.

**Distribution of MapDR.**    Figure 8 illustrates the diverse metadata distribution in the MapDR dataset. Subfigure (a) depicts the distribution of the time period for data collection, primarily from $07:00$ AM to $06:00$ PM, indicating that the dataset was mainly collected during daytime. Subfigure (b) displays the majority of clips containing between $30$ and $45$ frames.

**Auxiliary Evaluation Results.**    We conducted separate evaluations on all traffic signs of different lane types in MapDR. As shown in Table 4, the results indicate that the prediction difficulty varies among different categories of traffic signs.

Table 4: **Evaluation results of all traffic signs with different lane types in MapDR.** The results are all based on our method, and the split of dataset remains unchanged.

| **Metric** | BusLane | DirectionLane | EmergencyLane | VariableDirectionLane | |
|---|---|---|---|---|---|
| $P_{R.E.}(\%)$ | 73.44% | 78.44% | 92.20% | 71.42% | |
| $R_{R.E.}(\%)$ | 71.98% | 77.36% | 91.03% | 57.14% | |
| $P_{C.R.}(\%)$ | 73.34% | 82.12% | 92.85% | 71.42% | |
| $R_{C.R.}(\%)$ | 76.76% | 87.03% | 91.00% | 85.71% | |
| **Metric** | NonMotorizedLane | VehicleLane | TidalFlowLane | MultiLane | SpeedLimitedLane |
| $P_{R.E.}(\%)$ | 80.00% | 88.88% | 0% | 82.09% | 60.34% |
| $R_{R.E.}(\%)$ | 72.00% | 74.41% | 0% | 82.56% | 53.85% |
| $P_{C.R.}(\%)$ | 85.41% | 61.90% | 0% | 81.33% | 88.15% |
| $R_{C.R.}(\%)$ | 83.67% | 72.22% | 0% | 83.94% | 97.10% |

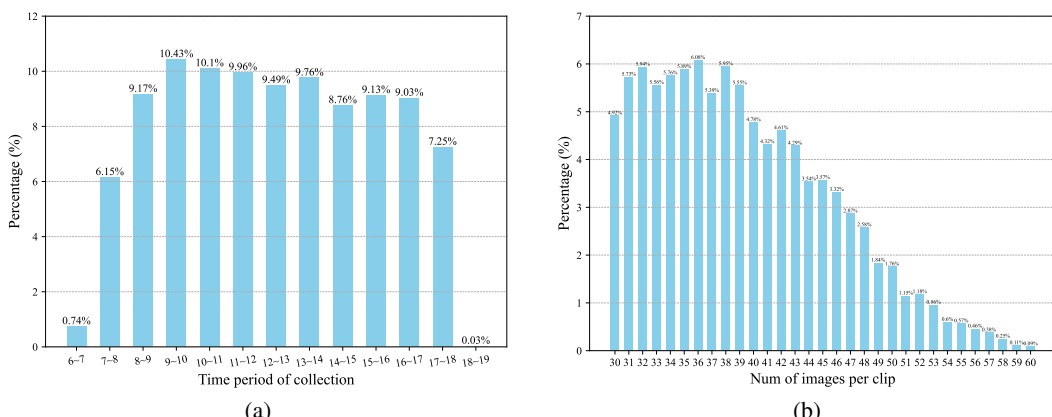

(a)                (b)

Figure 8: Distribution of MapDR.

**Potential negative societal impacts.** To minimize negative societal impact, we have applied obfuscation techniques to license plate numbers, facial features, and other personally identifiable information in our dataset. Additionally, sensitive geographical locations have been excluded, and coordinates in the ENU coordinate system have been provided without reference points to safeguard privacy. However, considering the potential inaccuracies and deviation of data distribution, the model may have misinterpretations and biases during the learning process. If such models are used on public roads, it could pose safety issues. Therefore, we recommend thorough testing of models before deploying to any autonomous driving system.

# H  VISUALIZATION OF MAPDR

Figure 11 visualizes driving rules for different lane types in the dataset, including BEV and front-view images, as well as formatted driving rules. The red pentagram in the BEV image marks the position of the traffic sign. The front-view image displays the lane vectors and manually annotated semantic groups, with driving rules organized as sets of $\{key : value\}$ pairs.

Figure 12 shows diverse types of traffic signs collected at different times, locations, and weather conditions, demonstrating rich inter-class differences and intra-class diversity, highlighting the complexity of the MapDR dataset.

# I    EXAMPLE FOR EVALUATION METRIC

We provide an example of metric calculation as Figure 9 shown, illustrating the evaluation process. Given the ground truth $G$ with 5 rule nodes and 8 centerline nodes while 6 edges between them, we assume that the algorithm has predicted $\hat{G}$ with 6 rules and 5 edges, the metric calculation process is detailed as below.

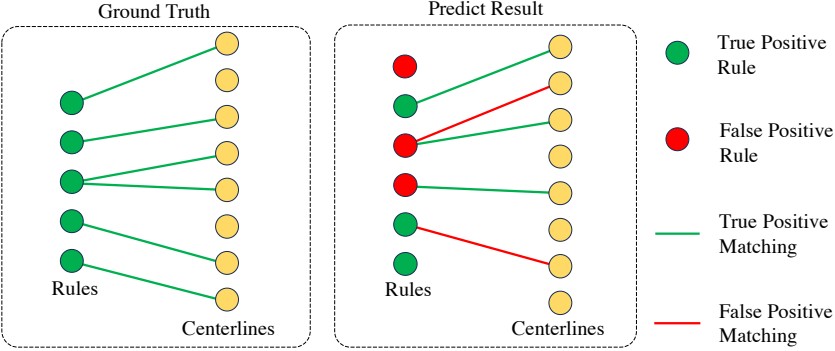

Figure 9: Illustration for Evaluation Metrics.

First, for the **Rule Extraction from Traffic Sign** sub-task, the ground truth has 5 rules, while the algorithm predicted 6 rules, of which 3 are correct (green circles) and 3 are incorrect (red circles). Then the precision ($P_{R.E.}$) and recall ($R_{R.E.}$) are calculated as Equation equation 4:

$$P_{R.E.} = \frac{|\hat{R} \cap R|}{|\hat{R}|} = \frac{3}{6} \qquad R_{R.E.} = \frac{|\hat{R} \cap R|}{|R|} = \frac{3}{5} \qquad (4)$$

Next, for the **Rule-Lane Correspondence Reasoning** task, there are 6 association results in the ground truth, but the algorithm predicted 5, with 3 being correct (green lines) and 2 being incorrect (red lines). Then, the precision ($P_{C.R.}$) and recall ($R_{C.R.}$) are calculated as Equation equation 5:

$$P_{C.R.} = \frac{|\hat{E} \cap E|}{|\hat{E}|} = \frac{3}{5} \qquad R_{C.R.} = \frac{|\hat{E} \cap E|}{|E|} = \frac{3}{6} \qquad (5)$$

Finally, considering the entire task, in the ground truth, a total of 6 lanes are assigned driving rules. The model predicted driving rules for 5 lanes, with correct predictions for both the association relationship and driving rules for only 1 lane. Therefore, the precision ($P_{all}$) and recall ($R_{all}$) for the entire task are calculated as Equation equation 6:

$$P_{all} = \frac{|\hat{G}^s \cap G^s|}{|\hat{G}^s|} = \frac{1}{5} \qquad R_{all} = \frac{|\hat{G}^s \cap G^s|}{|G^s|} = \frac{1}{6} \qquad (6)$$

# J    IMPLEMENTATION DETAILS

All experiments are conducted using PyTorch 1.8.0 on 8 NVIDIA V100 16G GPUs. We utilize pre-trained weights of DeiT Touvron et al. (2021) and BERT Devlin et al. (2019) to initialize the model in our experiments. Both of these assets are licensed under the Apache-2.0 license. Additionally, we have adopted ALBEF Li et al. (2021) as our code base, which is available under the BSD 3-Clause license.

## J.1 VISION-LANGUAGE ENCODER (VLE)

**Hyperparameters and Configurations.** We conduct $lr = 1e - 4$, $warmup\_lr = 1e - 5$, $decay\_rate = 1$, $weight\_decay = 0.02$, $embedding\_dim = 768$, $momentum = 0.995$, $alpha = 0.4$, $attention\_heads = 12$, and $batch\_size = 32$ for all experiments. We initialize vision encoder with pre-trained weight of DeiT Touvron et al. (2021), text encoder and fusion encoder with the first 6 layers and last 6 layers of BERT Devlin et al. (2019), respectively. The fine-tuning epoch is set to 50. Input image is resized to $256 \times 256$. The maximum number of tokens for input in the text encoder is 1000. *RandomAugment* is used, with hyperparameters $N = 2$, $M = 7$, and it includes the following data augmentations: *"Identity", "AutoContrast", "Equalize", "Brightness", "Sharpness"*.

**Clustering head.** We calculate the cosine similarity between the `[STC]` tokens to determine if they represent the same rule. The training procedure is supervised by *Contrastive Loss*. The positive margin is set to 0.7, and the negative margin is set to 0.3.

**Understanding head.** For properties in each rule, we prefer to classify their value into pre-defined classes. Specifically, for *"RuleIndex", "LaneType", "AllowedTransport", "EffectiveDate"* we employ linear layer to perform classification with *Cross-Entropy Loss*. For *"LaneDirection"*, this property is predicted by a multi-label classification that direction is defined as a combination of multi-choice from [*"None","Forbidden","GoStraight","TurnLeft","TurnRight","TurnAround"*]. The training loss is *Binary Cross-Entropy Loss*. Additionally, properties of *"EffectiveTime"*, *"LowSpeedLimit"* and *"HighSpeedLimit"* are formed as *string*. In practice, we classify the `[STC]` token to determine whether the OCR text is time or speed and use the original OCR text as the predicted value of these three properties.

## J.2 MAP ELEMENT ENCODER (MEE)

**Hyperparameters and Configurations.** We conduct $lr = 1e - 4$, $warmup\_lr = 1e - 5$, $decay\_rate = 1$, $weight\_decay = 0.02$, $embedding\_dim = 768$, $momentum = 0.995$, $alpha = 0.4$, $attention\_heads = 12$, and $batch\_size = 48$ for all experiments. We train MEE from scratch, the training epoch is set to 120. The maximum number of tokens for input in the vector encoder is 1000. The formatted rule is mapped to a 768-dimensional vector by an MLP. Specifically, each property in the rule is mapped to a 768-dimensional vector (except for *"EffectiveTime"*, *"LowSpeedLimit"* and *"HighSpeedLimit"*), and the position of the traffic sign is also mapped to a 768-dimensional vector through a position encoding method (as described in the main submission), and finally, all these vectors are added together to obtain the final feature of the rule. In MEE, there are a total of four types of embeddings: vector embedding, position embedding, type Embedding, and instance embedding. The encoding method for vector embedding and position Embedding is detailed in the main submission. For type embedding, as there are 5 types in total, we initialize it using $nn.Embedding$, with the hyperparameters $num\_embeddings = 5$ and $embedding\_dim = 768$. Similarly, we also use $nn.Embedding$ to initialize the instance embedding, with the $num\_embeddings = 120$ and $embedding\_dim = 768$, meaning it can support a maximum of 120 vectors. It is important to note that since the instance embedding is only used to distinguish different vectors, we shuffle the order of these embeddings at each iteration. After the multimodal fusion encoder of MEE, we further incorporate an $nn.Linear$ to map the 768-dimensional features to 256, which is then connected to the association head.

**Association head.** We perform binary classification on `[VEC]` tokens to determine whether the vector is corresponding to the input rule. The training procedure is supervised with *Binary Cross-Entropy Loss*.

## J.3 ANALYSIS OF EVALUATION ERROR

We conduct multiple experiments on our method with various random seed, and the experimental results are shown in Figure 10. We repeated all experiments 5 times with various seeds which are depicted in different colors. We uniformly sampled 100 points within the range of 0 to 1 as the binary classification threshold for association head in correspondence reasoning procedure, and then

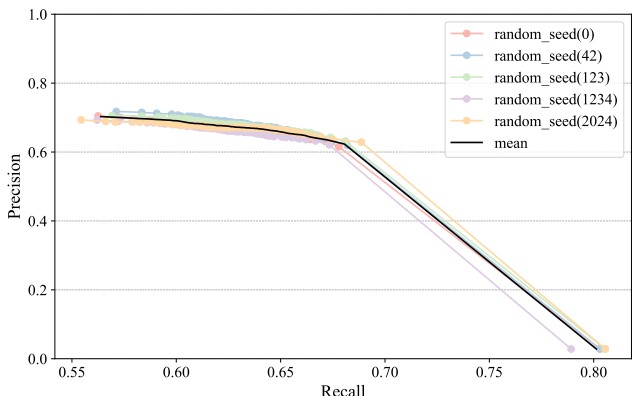

Figure 10: Overall P-R curves with various random seeds.

calculate the $P_{all}$ and $R_{all}$ for each threshold. The mean fitted line is shown in black, demonstrating the stability of our method. Specifically, we calculated the standard deviation of all evaluation metrics at a fixed threshold among different random seeds. For rule extraction sub-task, the standard deviation of $P_{R.E.}$ and $R_{R.E.}$ are 0.32 and 0.38. In the rule-lane correspondence reasoning sub-task the standard deviations are 0.07 and 0.38 for $P_{C.R.}$ and $R_{C.R.}$. Overall, the standard deviations of $P_{all}$, $R_{all}$ and $AP$ are 0.18 0.10 and 1.07, respectively.

## K QUALITATIVE RESULTS OF MLLM

We qualitatively evaluated the performance of existing MLLMs on the two subtasks of **Rule Extraction** and **Correspondence Reasoning** using a subset of MapDR, which consists of 20 randomly sampled examples for traffic signs among all lane types, totaling 180 cases. Annotators subjectively assessed the correctness of MLLM outputs. Since MLLMs cannot provide confidence scores for their predictions, we could not use a threshold to calculate precision and recall metrics. Therefore, we evaluated accuracy, specifically $Acc_{R.E.} = \frac{|\hat{R} \cap R|}{|R|}$ and $Acc_{C.R.} = \frac{|\hat{E} \cap E|}{|E|}$, as shown in Table 5.

Table 5: **Accuracy on the subset of MapDR.** MLLMs are subjectively evaluated by annotators, so the results only approximately reflect their capacity.

| Model | $Acc_{R.E.}(\%)$ | $Acc_{C.R.}(\%)$ |
|---|---|---|
| Qwen-VL Max Bai et al. (2023) | 44.4 | 20.6 |
| Gemini Pro Team et al. (2024) | 31.1 | 6.1 |
| Claude3 Opus Anthropic (2024) | 4.4 | 1.1 |
| GPT-4V OpenAI | 3.3 | 1.7 |
| Ours | 65.15 | 78.84 |

All existing MLLMs are evaluated without SFT, clearing former memories before each prompt to avoid contextual influence. This experiment primarily aims to qualitatively analyze the zero-shot capacity of MLLMs in traffic scene understanding, rather than a rigorous quantitative comparison. Overall, the results highlight the necessity of this task and dataset.

As all the traffic signs and rules are from China, described in Chinese, we utilized a Chinese prompt. In Figure 13, we present our input, including the image and prompt, along with the results generated by MLLMs. Our prompt can be translated as: *"What is the meaning of the traffic sign in the red box? In this picture, the red lines represent the lane centerlines, which centerline or centerlines are related to the traffic sign in the red box?"*. The use of a Chinese prompt may also contribute to Qwen-VL's better performance, as it originates from Alibaba, a Chinese company, and its training process involved more Chinese text compared to other models Bai et al. (2023).

Additionally, we referenced Shtedritski et al. (2023) to mark the red boxes and red lines in the images as visual prompts for the signs of interest and the centerlines of the lanes, which is convenient but may not be the most effective method and may also limit the performance of MLLMs. Furthermore, according to Rang et al. (2023), we can learn that apart from the Qwen-VL model, other models such as GPT-4V have weak capabilities in Chinese OCR, so this possibly limit their cognitive performance. Overall, despite MLLMs' zero-shot performance not achieving remarkable results, they possess significant potential. We believe that with further prompt optimization, the implementation of SFT, and other methods, larger models will undoubtedly achieve improved results in the future.

Table 6: Data Composition.

| Key | Subkey | Sub-subkey | Type | Value |
|---|---|---|---|---|
| "traffic_board_pose" | / | / | List[List[float]] | $[[x_1, y_1, z_1], \dots]$ |
| "vector" | "0" | "type" | Single Select | "0" (Divider)
"1" (Special Divider)
"2" (Road Boundary)
"3" (Centerline)
"4" (Crosswalk) |
| | | "vec_geo" | List[List[float]] | $[[x_1, y_1, z_1], \dots]$ |
| | $\cdots$ | | | |
| "camera_intrinsic_matrix" | / | / | List[List[float]] | $[[f_x, 0, c_x],$
$[\ 0, f_y, c_y],$
$[\ 0,\ 0,\ 1\ ]]$ |
| "camera_pose" | "{timestamp}" | "tvec_enu" | List[float] | $[t_1, t_2, t_3]$ |
| | | "rvec_enu" | List[float] | $[r_1, r_2, r_3, r_4]$ |

Table 7: Label Composition. "None" denotes the rule does not restrict the specific property. The property "LaneDirection" is represented by the combination of multiple selected basic directions.

| Key | Subkey | Sub-subkey | Type | Value |
|-----|--------|-----------|------|-------|
| "0" | "attr_info" | LaneType | Single Select | "DirectionLane"
"BusLane"
"EmergencyLane"
"VariableDirectionLane"
"Non-MotorizedLane"
"VehicleLane"
"TidalFlowLane"
"MultiLane"
"SpeedLimitedLane" |
| | | RuleIndex | Str | eg: "0" |
| | | LaneDirection | Multiple Select | "None", GoStraight"
"TurnLeft", "TurnRight",
"TurnAround", "Forbidden" |
| | | AllowedTransport | Single Select | "None"
"Bus"
"Vehicle"
"Non-Motor"
"Truck" |
| | | EffectiveDate | Single Select | "None"
"WorkDays |
| | | EffectiveTime | Str | eg: "7:00-9:00 " |
| | | LowSpeedLimit | Str | eg: "40" |
| | | HighSpeedLimit | Str | eg: "120" |
| | "centerline" | / | List[int] | eg: [16, . . . ] |
| | "semantic_polygon" | / | List[List[float]]] | $[[x_1, y_1, z_1], . . . ]$ |

. . .

Listing 1: Example of data file.

```
{
    "traffic_board_pose": [
        [6250.741478919514, -23002.897461687568, -51.60124124214053 ],
        [6250.767766343895, -23002.852551855587, -53.601367057301104],
        [6247.90629957122,  -23005.522309921853, -53.698920409195125],
        [6247.880012146425, -23005.5672197543,   -51.69879459403455 ]
    ],
    "vector": {
        "0": {
            "type": "2",
            "vec_geo": [
                [6222.740794670596, -22977.551953653423, -59.28851334284991 ],
                [6224.65054626556,  -22979.753116989126, -59.31985123641789 ],
                [6229.777790947785, -22985.886256590424, -59.40054347272962 ],
                [6237.236963539255, -22995.08138003234,  -59.51233040448278 ],
                [6242.709547414123, -23002.134314719562, -59.58363144751638 ],
                [6247.894389983971, -23008.135111707456, -59.648408086039126],
                [6253.242476279292, -23014.058069147195, -59.7004144266624775],
                [6258.56982873722,  -23020.026259167204, -59.72872495371848 ]
            ]
        },
        "1":{ ...... },
    "camera_intrinsic_matrix": [
        [904.9299114165748, 0.0,                949.2163397703193],
        [0.0,               904.9866120329268, 623.7475554790544],
        [0.0,               0.0,                1.0              ]
    ],
    "camera_pose": {
        "1710907374739989000": {
            "tvec_enu": [6217.6643413086995, -22963.182929283157, -57.714795432053506],
            "rvec_enu": [-0.2097012215148481, 0.6478309996572192,
                         -0.6804515437189796, 0.2707879063036554]
        },
    }
}
```

Listing 2: Example of label file.

```
{
    "0": {
        "attr_info": {
            "LaneType":         "DirectionLane",
            "RuleIndex":        "1",
            "LaneDirection":    ["GoStraight","TurnLeft"],
            "EffectiveTime":    "None",
            "AllowedTransport": "None",
            "EffectiveDate":    "None",
            "LowSpeedLimit":    "None",
            "HighSpeedLimit":   "None"
        },
        "centerline": [17],
        "semantic_polygon": [
            [6250.473053530053, -23003.147903473426, -51.91421646422327],
            [6250.387053162556, -23003.22814210385,  -53.56106227565867],
            [6249.308139461227, -23004.234772194584, -53.48654436563898],
            [6249.381109470012, -23004.166690932405, -51.82106907669865]
        ]
    },
    "1": {
        "attr_info": {
            "LaneType":         "DirectionLane",
            "RuleIndex":        "2",
            "LaneDirection":    ["GoStraight"],
            "EffectiveTime":    "None",
            "AllowedTransport": "None",
            "EffectiveDate":    "None",
            "LowSpeedLimit":    "None",
            "HighSpeedLimit":   "None"
        },
        "centerline": [16],
        "semantic_polygon": [
            [6249.081411219644,  -23004.446310402054, -53.45673720163109 ],
            [6249.21171480676,   -23004.324736719598, -51.76890653968486 ],
            [6248.1406193206585, -23005.324072389387, -51.694386629665156],
            [6248.0546189531615, -23005.404311019807, -53.37476750060943 ]
        ]
    }
}
```

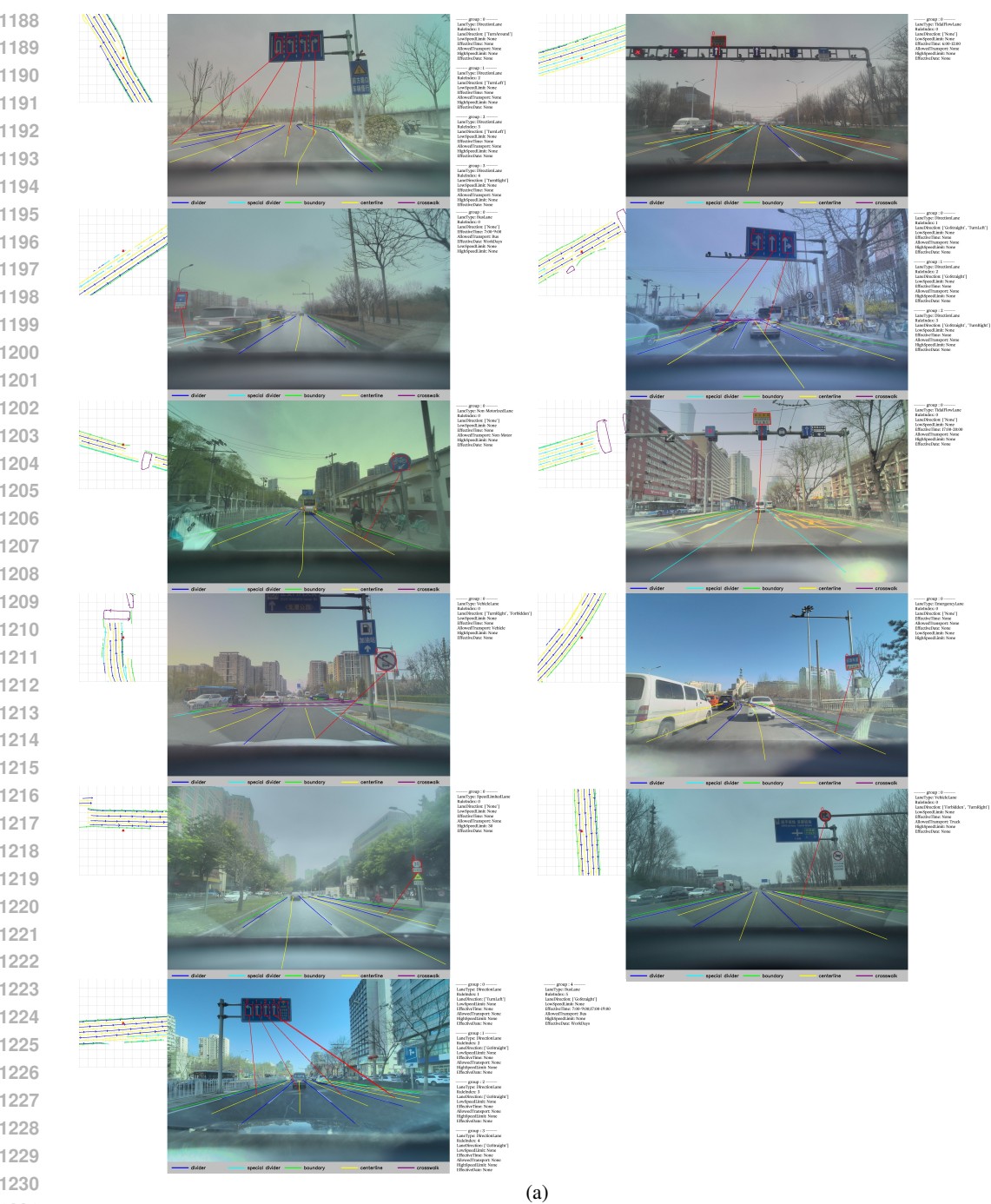

(a)

Figure 11: Visualization of MapDR.

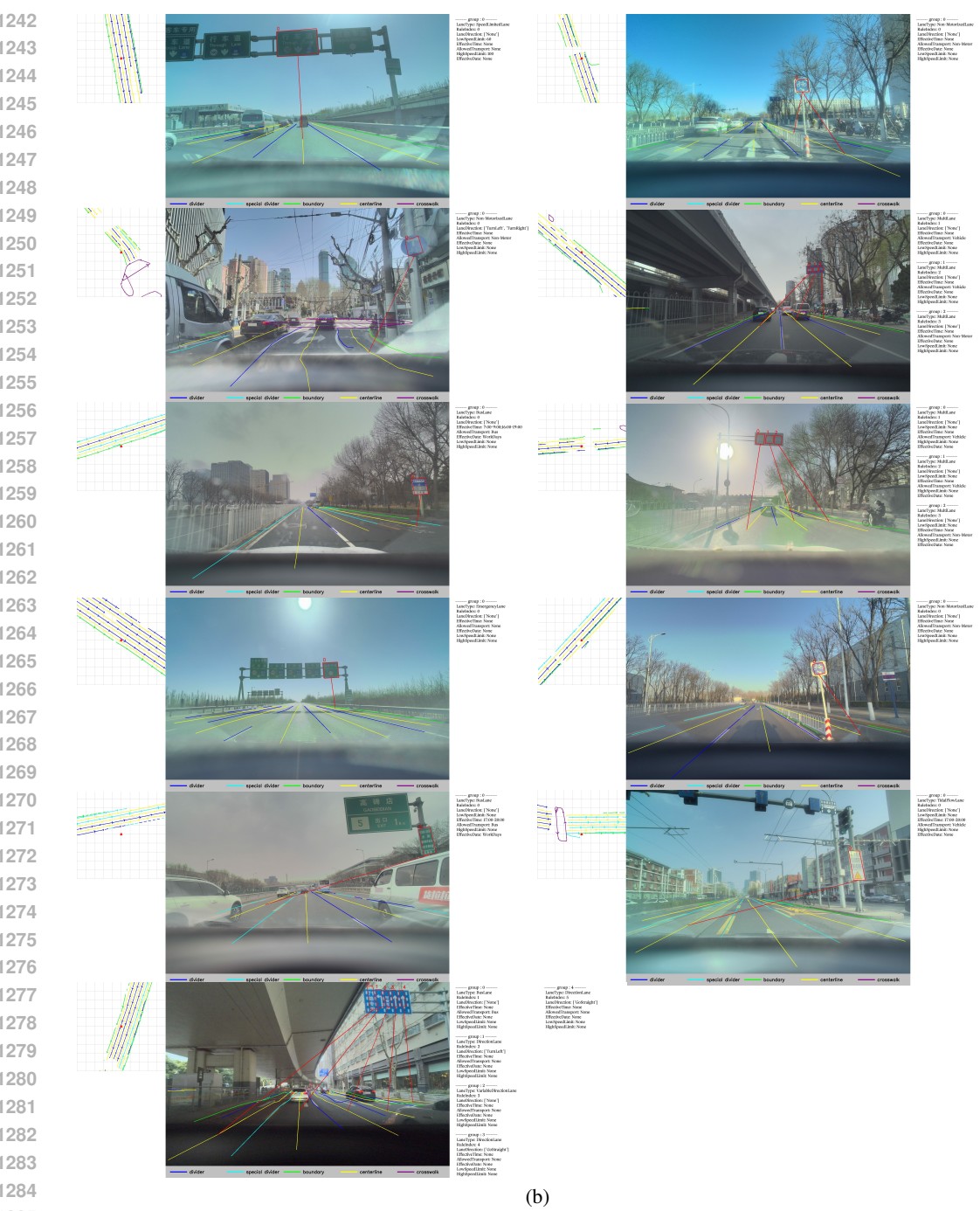

(b)

Figure 11: Visualization of MapDR.

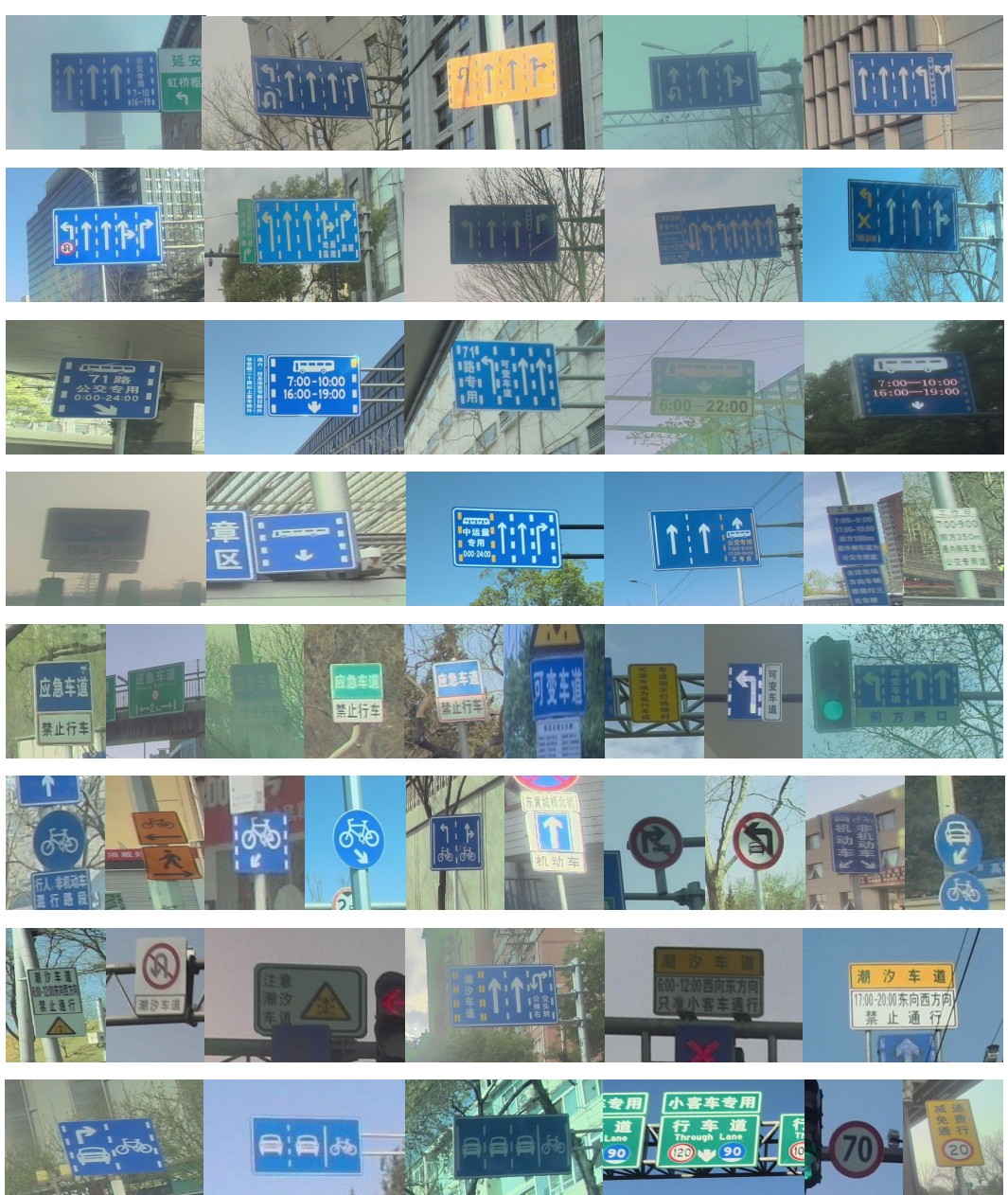

Figure 12: Visualization of traffic signs.

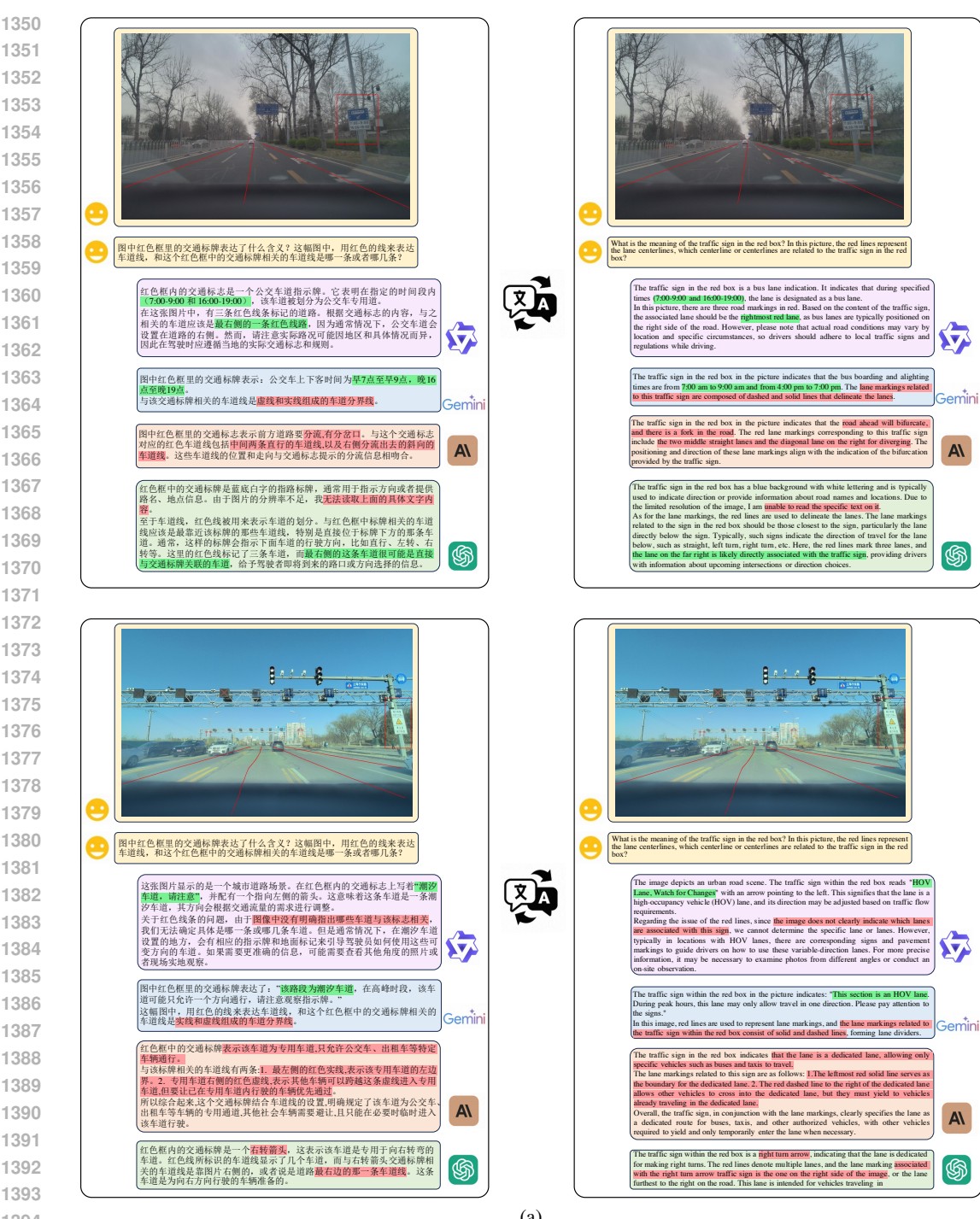

(a)

Figure 13: Prompts and answers for MLLMs.

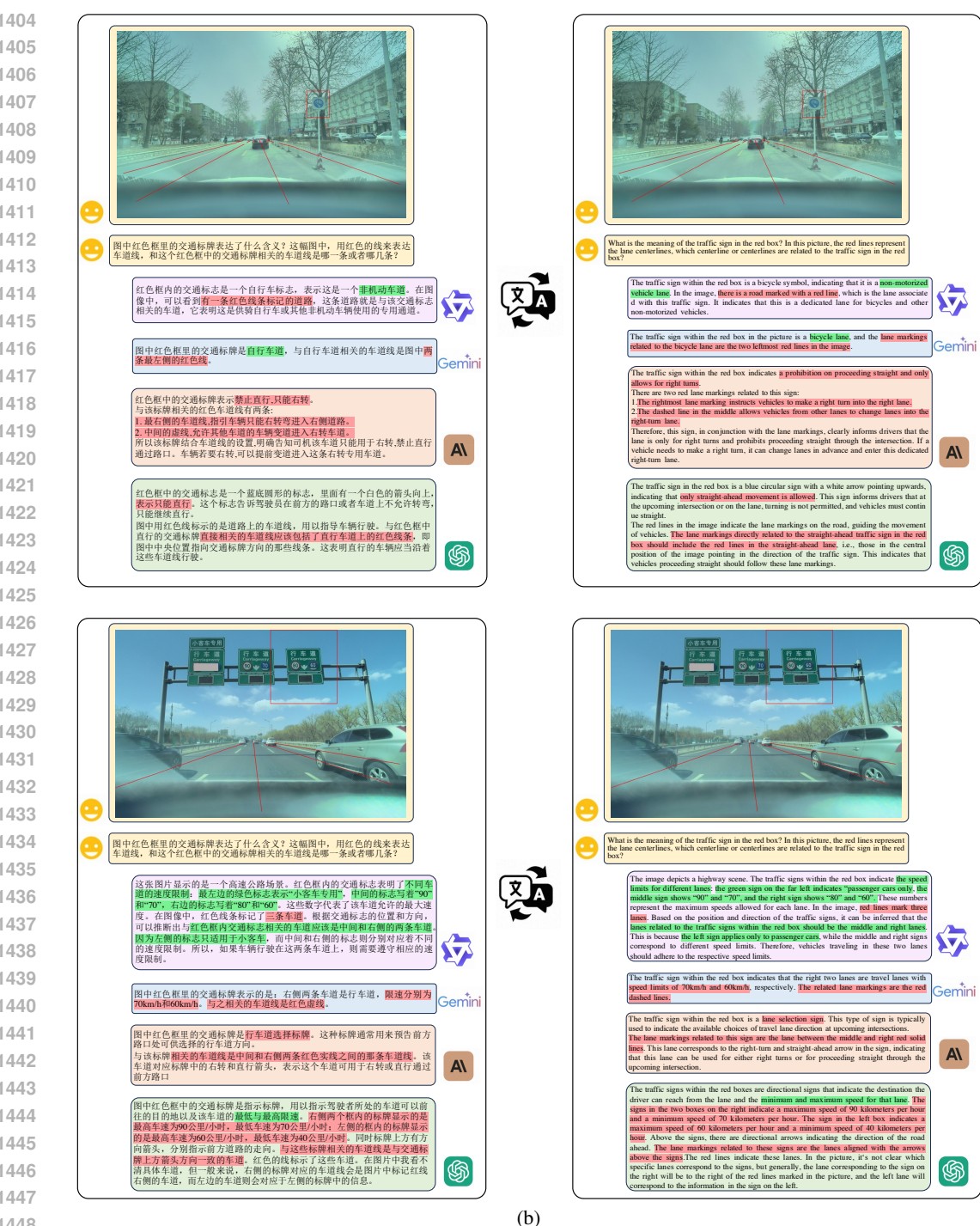

(b)

Figure 13: Prompts and answers of MLLMs.

