# OpenReview forum: "Driving by the Rules: A Benchmark for Integrating Traffic Sign Regulations into Vectorized HD Map"
_ICLR.cc/2025/Conference — ICLR 2025 Conference Withdrawn Submission_

### Official Review · Reviewer_XBRc · 2024-11-01

**Soundness:** 2
**Presentation:** 2
**Contribution:** 2
**Rating:** 5
**Confidence:** 3

**Summary:**

This paper proposes a new dataset MapDR for benchmarking autonomous navigation systems to follow traffic rules. MapDR associates driving rules extracted from traffic signs with vectorized HD maps. MapDR provides over 10000 video clips annotated with traffic sign regulations and their correspondences to lanes, enabling 2 new tasks: rule extraction from traffic signs and Rule-Lane Correspondence Reasoning. Rule extraction from traffic signs is to predict rules given images sequences as input and Rule-Lane Correspondence Reasoning is to associate lanes with all extracted rules. Based on this dataset and these new tasks, this paper also presents two baseline methods Vision-Language Encoder(VLE) and Map Element Encoder(MEE) for these sub-tasks. Experiments reflect the finetuned baseline methods can outperform their baselines and existing general-purpose Multi-modal LLM like GPT-4o and Qwen-VL Max.

**Strengths:**

1. The paper is well structured. The proposd dataset and benchmark contain large amount of traffic data with lane-based annotations.

**Weaknesses:**

1. The LLM-based autonomous driving works[1,2,3,4] are poorly mentioned in this submission. As the target of this dataset is estimate the whether autonomous driving models can correct actions given the visual input of lane-based rules, it is important to discuss this field in the related work and evaluate their performance with your benchmark.
2. There is poor comparison for the proposed baseline methods in this paper. The models are only compared to the base BERT and ALBEF models. No autonomous driving related works are compared.
3. There is no benchmark result for autonomous driving related works on their dataset.
4. As the target is to reason lane-based rules from visual inputs to perform correct action, the necessity to split this task to two sub-tasks is not discussed. It seems the two sub-tasks can be solved end-to-end. More justification for these two-sub tasks are expected.
5. The diversity of the proposed dataset seems to be limited. RS10K contains 10k images from 31 cities in China. However, MapDR in this work mainly covers Beijing and Shanghai in China.

[1] Zhenhua Xu, et.al. DriveGPT4: Interpretable End-to-end Autonomous Driving via Large Language Model. Arxiv 2023

[2] GPT-Driver: Learning to Drive with GPT. NeurIPS 2023 Workshop

[3] Driving with LLMs: Fusing Object-Level Vector Modality for Explainable Autonomous Driving. ICRA 2024

[2] DriveLM: Driving with Graph Visual Question Answering. ECCV 2024

**Questions:**

1. For experiments in Table 5, how does it perform if the prompt is in English and why do you use the same color for highlighting traffic sign and lanes? If using different color, will the performance be improved?
2. For Table 2, why the baseline methods cannot converge for Correspondence Reasoning task?

---

### Official Review · Reviewer_UjqL · 2024-11-02

**Soundness:** 2
**Presentation:** 3
**Contribution:** 3
**Rating:** 6
**Confidence:** 4

**Summary:**

**Key Findings**:

1. **Novel Dataset**: This paper introduces MapDR, a dataset for integrating traffic sign regulations into HD maps for autonomous driving, containing over 10,000 annotated videos and 18,000 lane-specific driving rules.

2. **Autonomous Navigation Sub-Tasks**: MapDR defines two sub-tasks—(a) extracting rules from traffic signs, and (b) linking these rules to specific lanes on HD maps.

3. **Baseline Model**: A multimodal baseline model with Vision-Language and Map Element Encoders is proposed, effectively integrating images, text, and vectors to setup a baseline for the two sub-tasks.

4. **Dataset Utility**: MapDR enhances rule contextualization and lane mapping, outperforming existing datasets by improving rule interpretation for autonomous vehicles.

---

These findings underscore MapDR's potential to extend current methods on HD map reconstruction, advancing autonomous vehicle development.

**Strengths:**

Strengths

1. This work is sufficient in its contribution, including a newly-collected dataset with novel annotations to fill the gaps between HD map reconstruction and traffic scenario understanding. Meanwhile a baseline method combining a VLM and a map element encoding network is proposed, setting a baseline for the proposed dataset.
2. The writing is fluent and the figures are illustrative. The related work covers necessary related topics such as different HD map datasets and their pros and cons. And the figures are illustrative at an idea level, especially the visualization of the method pipeline.
3. The dataset annotation process is thorough and the dataset scale is sufficient for research idea validation.

**Weaknesses:**

Weakness

1. Motivation. Current navigation maps such as Google maps or Apple maps already have lane-level traffic rules adhering to ego-vehicles position. To be specific, in the figure 1, information such as driving direction per lane (go straight or turn left), speed limitation, and bus lane effective data are already available in the navigation maps like Google maps. Could the author discuss in detail about the necessity of the task 1 and 2 stated in this paper? This is my biggest concern about this paper.
2. Key difference to OpenLane-V2. In Table 1, the main difference to OpenLane-V2 seems to be the formatted driving rules. While in OpenLane-V2 there are also some annotations about driving directions. To be specific, it would be great if the author could provide a more detailed comparison between MapDR and OpenLane-V2, highlighting specific differences in the formatted driving rules, level of detail, or types of rules captured, and explain how these differences translate to practical advantages for autonomous driving applications.

**Questions:**

1. Any challenging scenarios? The demo in figure 3 is not quite difficult to convince the general audience that this direction worths exploration.
2. Is there any city-level difference discovered in the annotation, although the three cities are all from China?
3. Specific questions on motivation:
- How their approach differs from or improves upon existing navigation map solutions, particularly for autonomous driving applications
- What specific advantages their method offers that current maps do not provide, especially in real-time or dynamic scenarios
- Whether there are any limitations of current navigation maps in terms of integration with autonomous systems that their approach addresses

---

### Official Review · Reviewer_SoW2 · 2024-11-03

**Soundness:** 2
**Presentation:** 2
**Contribution:** 2
**Rating:** 3
**Confidence:** 5

**Summary:**

This paper presents MapDR, a novel dataset designed for extracting driving rules from traffic signs and associating them with vectorized, locally perceived high-definition (HD) maps. MapDR includes over 10,000 annotated video clips that capture the complex relationship between traffic sign regulations and lane information. The authors propose a multimodal approach that establishes a strong baseline for lane-level traffic scene understanding.

**Strengths:**

1. To the best of my knowledge, this is the first work to explore the correlation between traffic signs and lanes based on visual inputs, as shown in Figure 3. The task becomes highly complex when these elements must be associated, presenting a significant challenge for the computer vision community.

2. Labeling *MapDR* is a non-trivial effort, as it involves tagging a diverse set of attributes across more than 10,000 annotated video clips, 400,000 images, and 18,000 driving rules—requiring a substantial amount of work.

3. The two tasks—extracting rules from signs and reasoning about rule-lane correspondence—are highly challenging and critical for advancing automated driving technologies.

**Weaknesses:**

1. **Unclear Motivation for the Proposed Task:** If the primary goal of this work is to enhance the efficiency of HD map construction, as suggested by Figure 6, then using HD maps as input makes sense. However, if this is indeed the goal, the need for the proposed dataset and method remains unclear. What are the limitations of using human annotators to label and associate traffic rules with lanes? The authors should clarify the motivation to strengthen the case for this work.

2. **Ambiguity in Problem Formulation:** The introduction frames the work from the perspective of visual traffic scene understanding, which involves building a visual model that predicts road structures, identifies lanes and traffic signs, and links traffic sign semantics with the corresponding lanes. If this is the motivation, then I agree that *MapDR* and the proposed task represent a valuable and challenging contribution to the community. In this scenario, the proposed framework at inference should not include vectorized maps as input. Therefore, the motivation and formulation of the proposed task require clarification, as it may necessitate a different set of experiments altogether.

3. **Justification for Constructing a New Dataset:** The authors should clarify the rationale for collecting a new dataset specifically in Shanghai and Beijing. For example, they might demonstrate that existing datasets lack sufficient traffic sign data, which limits the study of tasks such as rule extraction from signs and rule-lane correspondence reasoning. Providing detailed statistics on the limitations of existing datasets could help underscore the necessity of collecting new data.

4. **Ensuring Annotation Quality:** The authors should detail the methods used to ensure annotation quality. For instance, how many reviewers verified the accuracy of labeled rules? Without these specifics, it is difficult to assess the dataset’s quality and, consequently, the contribution of this work, given that dataset creation is a primary contribution.

5. **Need for a Simpler Baseline than VLM:** Establishing a rule-based baseline could justify the necessity of using Vision-Language Models (VLMs) for this task. For instance, one might first identify lanes and map attributes from HD maps, then use OCR outputs to establish rule correspondences. Evaluating this rule-based approach could provide insights into the complexity of the proposed task and determine whether simpler methods might suffice, reducing the need for more complex models like VLMs.

**Questions:**

**Motivation of the Proposed Task:**
•	What is the primary motivation behind this work? Is it to improve the efficiency of HD map construction?
•	If so, why is the proposed dataset and method necessary, rather than relying on human annotators to label and associate traffic rules with lanes?
•	Can the authors clarify any specific limitations or bottlenecks with human labeling that the dataset aims to address?

**Problem Formulation:**
•	Could the authors clarify whether the intended goal is to enhance visual traffic scene understanding or HD map construction?
•	If the motivation is visual scene understanding, why are vectorized maps included as input during inference?
•	Could the authors explain how the problem formulation might affect the choice of experiments or required dataset features?

**Justification for a New Dataset:**
•	Why was it necessary to collect a new dataset in Shanghai and Beijing?
•	Do existing datasets lack sufficient traffic sign data for tasks such as rule extraction and rule-lane correspondence reasoning?
•	Could the authors provide specific statistics or limitations in current datasets to justify the need for new data collection?

**Annotation Quality Assurance:**
•	How did the authors ensure the quality of the annotations in *MapDR*?
•	How many reviewers validated the labeled rules, and what quality control measures were used?
•	Given that the dataset’s primary contribution hinges on accurate labeling, could the authors describe the quality control process in detail?

**Consideration of a Simpler Baseline than VLM:**
•	Have the authors considered a simpler rule-based baseline as a comparison to VLMs?
•	For example, could a rule-based method using OCR outputs and HD map attributes be implemented as a benchmark?
•	If a simpler approach could handle the task, what specific complexities justify the need for more complex models like VLMs?

---

### Official Review · Reviewer_yGQ9 · 2024-11-04

**Soundness:** 2
**Presentation:** 3
**Contribution:** 3
**Rating:** 6
**Confidence:** 3

**Summary:**

This paper proposes MapDR, a large-scale and diverse dataset designed for the extraction of driving rules from traffic signs and their association with vectorized, locally perceived maps. Within MapDR, two sub-tasks are defined: rule extraction from traffic signs and rule-lane correspondence reasoning, to align rules from traffic signs with their respective lanes. The paper also provides a baseline consisting of a vision-language encoder (VLE) and a map element encoder (MEE) to solve the two sub-tasks, respectively.

**Strengths:**

1. The motivation of this paper is strong; with the progress of online map construction, matching traffic rules with maps remains an open research problem.

2. Decomposing the integration of traffic sign regulations with maps into two sub-tasks: rule extraction and correspondence reasoning, and formulating these as node set, edge set, and overall subgraph set predictions is well-justified.

3. The proposed baseline is effective; ablations demonstrate that each module benefits performance in covering the two sub-tasks.

4. The paper is well-written, with a clear flow from the motivation, to the definition of the task, the proposed dataset with its benchmark, and a corresponding solution. Examples or detailed descriptions are provided to illustrate the key points in the paper, enhancing the reader's understanding.

**Weaknesses:**

1. The main contribution of the proposed MapDR dataset over existing datasets, such as OpenlaneV2, is the formatted rules. However, the paper mentions the comparison on this point briefly. Please provide more discussion on why this contribution is significantly beneficial for autonomous driving.

2. The dataset lacks nighttime scenarios, which can significantly affect the visual effect. It would be beneficial to include more tidal flow lane scenarios because tidal flow lane rules are important for traffic compliance.

**Questions:**

Please address the weaknesses.

---

### Official Review · Reviewer_fga5 · 2024-11-04

**Soundness:** 2
**Presentation:** 3
**Contribution:** 2
**Rating:** 5
**Confidence:** 3

**Summary:**

The authors propose a new dataset for driving understanding that focuses on rule inference and rule-lane association. The dataset is formulated as extraction of per-lane rule extraction, and covers 10k driving scenes and 18k lane-rule examples, totalling 400k images.
The authors suggest precision and recall metrics for the rules, lanes, and overall associations as basic measures for algorithmic success over this dataset.

The authors propose a baseline solution for this using a 2-stage VLMs augmented by an OCR, and connected through an emission head to a traffic rule extractor. The baseline model is contrasted with simple applications of  large-scale VLMs that fail to a certain degree, while the baseline model obtains significantly better performance.

**Strengths:**

- The paper addresses an important problem. The authors propose a full pipeline in terms of data, algorithm, and evaluation for the problem.
- The paper goes beyond just English-speaking locales, which is important if we want to deploy AD/ADAS systems in the real world.
- The dataset has a good dense coverage in terms of driving scenes over the deployment sites.
- The paper is overall clear, and aided by a detailed appendix and example data.

**Weaknesses:**

- Having a more diverse set of locations, and corresponding rule sets would greatly enhance the relevance of this dataset. Specifically,
- The paper would benefit from pointing to the definitions of the rules, and how they are established, as in the paper and in the supplied dataset they are often referred to with an index, but no specification which will allow to determine whether the set of rules is comprehensive. How are they considered a full set / how specific are they? How complete is the coverage of rules over the environment? It’s hard to tell.
- How are the rule extraction handling, e.g. intersections of 2 lanes or more (e.g. merges or roundabout entries)? Hard to say.
- It seems the dataset only allows image-to-rule/lane detection, but ignores the non-visual association question of match vehicle behaviors to rules, which is an important cue, especially since the ultimate goal of rule detection is to enable planning and reasoning in AD/ADAS systems. While it seems like the authors had a localization system running, but did not use detection and tracking algorithm, even an imperfect solution there would have allowed a much broader set of algorithms to be tested.
- Would be good if the authors better justified and ablated possible architectures, such as the use VLM vs OCR addition, or the pipeline of (VLM+OCR) -> MEE -> output, as opposed to other choices (such as joint inference, or iterative updates) or shed light on some of the other design choices. As a dataset paper that explores a pretty diverse phenomena (traffic rules), the dataset is relatively limited, which raises the bar on the algorithm, even if it’s considered a baseline for the problem.
- How were the 3 locations selected? For a dataset that is examining extraction of human behavior, the 3 location being in China, and Chinese signs, etc, may limit the reproducibility of the results (cf. the author’s claim that only Qwen got reasonable results from all the LLMs tested), would similar data be possibly annotated on existing datasets that have English semantics, or other rules? The 3 locations feel like they are very specific deployment sites, rather than a full attempt for a dataset, but I might be wrong.


Additional minor comments include:
- L281 - “sever” -> “server”
- L300 - E \in {0,1}^{|R|x|L|} should inclusion, not “\in”
- L969 - “seed” -> “seeds”
- Would be good to cover other approaches for rules in driving, e.g. logic-based approaches, possibly as an additional subsection in the related works, with possible example works such as “Liability, Ethics, and Culture-Aware Behavior Specification using Rulebooks”, "Vehicle Trajectory Prediction Using Generative Adversarial Network With Temporal Logic Syntax Tree Features", “Formal methods to comply with rules of the road in autonomous driving: State of the art and grand challenges”, “Formalization of Interstate Traffic Rules in Temporal Logic”, and others.

**Questions:**

- Can you write the formatted rule’s schema instead of just referring to Fig 3?
- Can the authors relate to the stages in figure 6 in the main text
- Can you justify the choice of DEIT in L914?
- Would be good to quantify the number of scenes compared to non-vision driving behavior datasets, as these contain significantly higher numbers of scenes, cf. Argoverse2 or Waymo’s prediction datasets. Also, add a distribution of the travel time and distance. The length of the scenes can be estimated from the number of frames and  limit on the sample rate that is mentioned in L698, but it’s better to be specific.

**Details Of Ethics Concerns:**

More of a clarification - in L737 the authors state that "Due to the sensitive nature of the dataset, which involves geographical location information, the full dataset is under review FOR NOW, and will be released in the camera-ready version". The dataset can have randomly shifted coordinates to avoid localization without effort, as a way to reduce the problem, and the coordinates given in the subset are already with respect to some local frame. The authors should clarify how they intend to release the final dataset.

---

### Note · Authors · 2024-11-14

I have read and agree with the venue's withdrawal policy on behalf of myself and my co-authors.